# Anthocyanin Synthesis and the Expression Patterns of bHLH Transcription Factor Family during Development of the Chinese Jujube Fruit (*Ziziphus jujuba* Mill.)

**Qianqian Shi [1,2], Xi Li [1,2], Jiangtao Du [1,2] and Xingang Li [1,2,3,*]**

[1] College of Forestry, Northwest Agriculture and Forestry University, Yangling District, Xianyang 712100, China; shiqq@nwafu.edu.cn (Q.S.); lixi@nwafu.edu.cn (X.L.); just5forever@163.com (J.D.)

[2] Key Comprehensive Laboratory of Forestry of Shaanxi Province, Northwest A&F University, Yangling District, Xianyang 712100, China

[3] Research Center for Jujube Engineering and Technology of State Forestry Administration, Northwest A&F University, Yangling District, Xianyang 712100, China

* Correspondence: xingangle@nwsuaf.edu.cn; Tel.: +86-29-87082556

**Abstract:** The basic helix–loop–helix (bHLH) family is an important transcription factor for eukaryotes and is involved in a wide range of biological activities. Among these, bHLH can interaction with WD repeat (WD40 or WDR) and V-myb avian myeloblastosis viral oncogene homolog (MYB) form a ternary complex to promote the efficient synthesis of anthocyanins. In this study, a total of 138 jujube bHLH (*ZjbHLH*) family members were screened from the transcriptome of the two jujube cultivars, 'Junzao' (JZ) and 'Tailihong' (TLH). Of these, 95 *ZjbHLH* genes were mapped to 12 chromosomes. A phylogenetic tree was constructed using 27 arabidopsis bHLH (*AtbHLH*) protein sequences of *Arabidopsis thaliana* (L.) Heynh. and 138 *ZjbHLH* protein sequences of jujube. The results show that the *ZjbHLH* family of jujube can be divided into 12 subfamilies. The three candidate genes, *ZjGL3a*, *ZjGL3b* and *ZjTT8*, related to anthocyanin synthesis, were classified into subgroup III. Meanwhile, *ZjGL3a*, *ZjGL3b* and *ZjTT8* have high homology with the bHLH transcription factors involved in anthocyanin synthesis in other plants. In addition, it was found that the jujube *ZjbHLH* transcript family showed changing patterns of expression during fruit development. The relative expression levels of *ZjGL3a*, *ZjGL3* and *ZjTT8* were consistent with the changes of the anthocyanin contents in the two jujube cultivars examined. To better understand the anthocyanin synthesis pathway involved in *ZjbHLH*, a regulatory pathway model for anthocyanin synthesis was constructed. This model involves the processes of anthocyanin signal transduction, synthesis and transport.

**Keywords:** *ZjbHLH* transcription factor; jujube; anthocyanin; anthocyanin synthesis pathway

## 1. Introduction

Chinese jujube (*Ziziphus jujuba* Mill.) belongs to the Rhamnaceae and is indigenous to China. It has been domesticated in China as a fruit crop for more than 7000 years [1]. This species is in high demand, due to its rich flavor and high nutritional value, in many Asian countries [2]. In more recent years, human nutritional science has demonstrated that jujube is a rich source of vitamins, flavonoids, polyphenols, fiber, sugars and other beneficial components [3]. Anthocyanins are widely distributed in the fruits, flowers, seeds and other tissues of the jujube plant. Here, they serve as antioxidants, attract insect pollinators, resist ultraviolet rays and participate in the plant's responses to biotic and abiotic stresses [4]. The synthesis of anthocyanins is closely related to the flavonoid synthesis pathway. Sequence studies of different plant species have shown that the anthocyanin synthesis gene is strongly conserved, and also that the flavonoid synthesis gene is especially conservative [5].

The basic helix–loop–helix (bHLH) family of transcription factors (named for its basic-helix-loop-helix structure) is large and widely distributed in eukaryotes. Its core conserved domain contains approximately 60 amino acid residues [6]. The bHLH has been shown to be involved in numerous biological processes, including stress resistance [7–9], growth and development [10,11], signal transduction [12,13] and synthesis [14–16]. The first bHLH transcription factor was discovered in maize, where its function is involved in the synthesis of anthocyanins [17]. Other family members have subsequently been found in several species, including *Arabidopsis* [18], tobacco [19], rice [20] and tomato [21]. At this stage, 150 arabidopsis bHLH (*AtbHLH*) family members have been identified from *Arabidopsis* and have been shown to be involved in a range of processes, including the development of the reproductive system, transduction of phytochrome signaling, synthesis of secondary metabolites, and stress responses [22]. In addition, bHLH plays an important role in the synthesis of anthocyanins [23,24], and bHLH and myeloblastosis viral oncogene homolog (MYB) transcription factors can promote pigment synthesis in *Arabidopsis* seedlings [25]. In apples, the *MdbHLH3* transcription factor has also been shown to promote anthocyanin accumulation in fruits [26]. The WD repeat (WD40)/basic helix–loop–helix (bHLH)/V-myb avian myeloblastosis viral oncogene homolog (MYB) complex activates downstream signaling cascade under jasmonic acid (JA) induction, thereby regulating anthocyanin accumulation [27]. In addition, bHLH also plays an essential role in the synthesis of anthocyanins in gentian flowers [28].

In the color mutant jujube cultivar 'Tailihong' (TLH), the red color is due to anthocyanins [29]. Many studies have shown that bHLH transcription factors are involved in numerous physiological and biochemical processes in other plants. However, the molecular mechanisms of pigment formation in jujube fruit have not yet been elucidated, nor has the bHLH transcription factor family of jujube been fully identified. Therefore, our aim was to identify and analyze the bHLH transcription factors in the jujube genome and to investigate their changing expressions during fruit development. Our aim is that these findings will deepen our understanding of the functions of the jujube bHLH (*ZjbHLH*) genes in regulating the anthocyanin synthesis mechanism in jujube fruit.

## 2. Materials and Methods

### 2.1. Materials

Two jujube cultivars, *Ziziphus jujuba* Mill. '*Junzao*' (JZ) and '*Tailihong*' (TLH), were obtained from the Jujube Experimental Station of Northwest Agriculture and Forestry University in Qingjian, Shaanxi, China. A total of 20 fruit samples of each genotype from three jujube trees were harvested at six developmental stages (S1 . . . S6) on days 30, 50, 80, 90, 100 and 110 after anthesis (DAA), respectively (see Figure S1). The fruit skins (about 2 mm thick) were removed with a domestic vegetable peeler randomly. Composite skin samples were immediately frozen in liquid nitrogen and held at −80 °C pending analysis.

### 2.2. Extraction and Analyses of Total Anthocyanins

Samples (about 0.5 g) of jujube fruit skin were extracted with 5 mL of 0.1% HCl in methanol for 24 h, at 4 °C in the dark. After centrifuging at 12,000 rpm for 15 min, three replicates were used for each sample [29]. The total anthocyanin content (TAC) was measured by the pH differential method. Absorbance was measured at 510 and 700 nm in buffers at pH 1.0 and at 4.5. The TAC is expressed as cyanidin-3-glucoside equivalents.

### 2.3. Total RNA Extraction and cDNA Synthesis

The total RNA of all samples was extracted using the TaKaRa MiniBEST Plant RNA Extraction Kit (TaKaRa, Beijing, China) according to the manufacturer's instructions. The purity and concentration of the extracted total RNA were determined using NanoDrop 20000 (Thermo Scientific, Pittsburgh, PA, USA). Samples with an OD260/280 ratio of 1.8 to 2.0 and an OD260/230 ratio greater than 2.0 were retained. Subsequently, total RNA was reverse transcribed to obtain first strand cDNA.

## 2.4. Real-Time Quantitative Polymerase Chain Reaction (RT-qPCR) Analysis

Primers for real-time Quantitative PCR (RT-qPCR) were designed using Primer Premier 7.0 software (Premier, Palo Alto, CA, USA) (Table 1). The relative expression levels of target genes were detected using total RNA from different developmental stages of the two jujube cultivars as templates in CFX96 Real-Time PCR Detection System (Bio-Rad, Hercules, CA, USA). *ZjUBQ* and *ZjUBQ2* were used as reference genes to correct the data [30]. The reaction system was 10 μL, including 1 μL cDNA, 1 μL each of the upstream and downstream primers, 5 μL of 2 × SYBR Premix Ex Taq II (TaKaRa) and 2 μL of ddH$_2$O. The PCR reaction procedure was 95 °C for 10 s, 58 °C for 30 s, 72 °C for 45 s, 35 cycles of the above three steps and 72 °C for 5 min to complete the reaction.

**Table 1.** RT-qPCR primers.

| Gene Name | F (Primer Sequence (5′-3′)) | R (Primer Sequence (5′-3′)) | Length (bp) |
|:---:|:---:|:---:|:---:|
| *ZjGL3a* | GCATTCTGCTGCATTGTCTC | CCCCTTTTTGCCTTTATTTT | 194 |
| *ZjGL3b* | CAGCCACACCCAACCACTA | CACCACACCTCCCAGAAAG | 279 |
| *ZjTT8* | ATCATCACACCCGCACAGAA | CCCAACCAAAAGAGAACCCA | 114 |
| *ZjUBQ* | TGGATGATTCTGGCAAAG | GTAATGGCGGTCAAAGTG | 98 |
| *ZjUBQ2* | CACCCGTTACTTGCTTTC | CTCTTCCCATTGTCCTCC | 93 |

## 2.5. Sequence Bioinformatics Analysis

Phylogenetic tree analysis of the bHLH transcription factor family obtained was generated using MEGA 7.0 (Center for Evolutionary Medicine and Informatics, Tempe, AZ, USA). The conserved domain of the bHLH transcription factor family protein sequence was analyzed using DNAMAN 8.0 (Lynnon Biosoft, San Ramon, CA, USA) and the online website WebLogo 3 (http://weblogo.berkeley.edu/logo.cgi). At the same time, their motif domains were predicted through the meme-suite website (http://meme-suite.org/). Chromosome localization of jujube *ZjbHLH* transcription factors was carried out using MapInspect 1.0 software (https://mapinspect.software.informer.com/1.0/). Analysis of the tertiary structure of the anthocyanin synthesis-related transcription factor protein was carried out using the online website, phyre2 (http://www.sbg.bio.ic.ac.uk/phyre2/html/page.cgibid=index).

## 2.6. Analysis of ZjbHLH Gene Expression from RNA-Seq Data

RNA-Seq reads were obtained with an Illumina HiSeq 2000 (Illumina, San Diego, CA, USA). The fragments per kilobase of exon per million mapped reads (FPKM) values were calculated based on RNA-Seq reads. The heatmap was generated with TBTOOLS software (https://github.com/CJ-Chen/TBtools) [31]. The color scale shown represents FPKM counts, and the ratios were log2 transformed.

## 3. Results

### 3.1. Analysis of Biological Information of the Jujube bHLH Transcription Factor Family

A total of 145 *ZjbHLH* transcriptome sequences were screened from the JZ genome and the sequence was analyzed by the Pfam online site (http://pfam.xfam.org/) to remove the incomplete sequence of the conserved domain. Finally, 138 *ZjbHLH* transcription factor family members were obtained. The amino acid sequences of the bHLH transcription factors of jujube and *Arabidopsis thaliana* were analyzed and classified using the MEGA 7.0 software with the bootstrap values from 1000 replicates. The *ZjbHLH* genes were divided into 12 subfamilies (Figure 1A) [32]. Among these, Group II and Group VI members contained only two *ZjbHLH* transcription factors, while Group V contains 30 *ZjbHLH* members. Three candidate *ZjbHLH* transcription factors (TFs) were found to belong to Group III and are highly correlated with anthocyanin synthesis (*ZjGL3a*, *ZjGL3b* and *ZjTT8*). To provide additional support for this selection, phylogenetic analyses indicated that *ZjTT8*, *ZjGL3a* and *ZjGL3b* belong to the anthocyanin identified in other plants (Figure 1B). The protein sequences of

*ZjTT8* are most closely related to *MabHLH3* and *FabHLH3*, sharing 78% and 71% amino acid identity, respectively. *ZjGL3b* showed the highest amino acid sequence identity with PhJAF13 (50%), while *ZjGL3a* showed the highest amino acid sequence identity with *MabHLH33* (67%). *MdbHLH3*, *PhJAF13* and *MdbHLH3* have been shown to interact physically with MYB TFs to regulate several structural genes involved in the synthesis of anthocyanin [4,26,33]. Figure 1C shows the gene IDs and physiological and biochemical characteristics of the three candidate genes (*ZjTT8*, *ZjGL3a* and *ZjGL3b*). The lengths of the protein sequences of *ZjTT8*, *ZjGL3a* and *ZjGL3b* are 642, 637 and 705 amino acids, respectively. Their molecular weights (Mw) are 72694.97, 70922.78 and 78688.7 Da, respectively. Their isoelectric point (pI) values are 5.63, 5.19 and 5.22, respectively. Their aliphatic index (Ai) is 79.89, 84.65 and 74.16, respectively. The grand average values of hydropathicity (GRAVY) of the candidate *ZjbHLHs* were −0.545, −0.478 and −0.632, respectively.

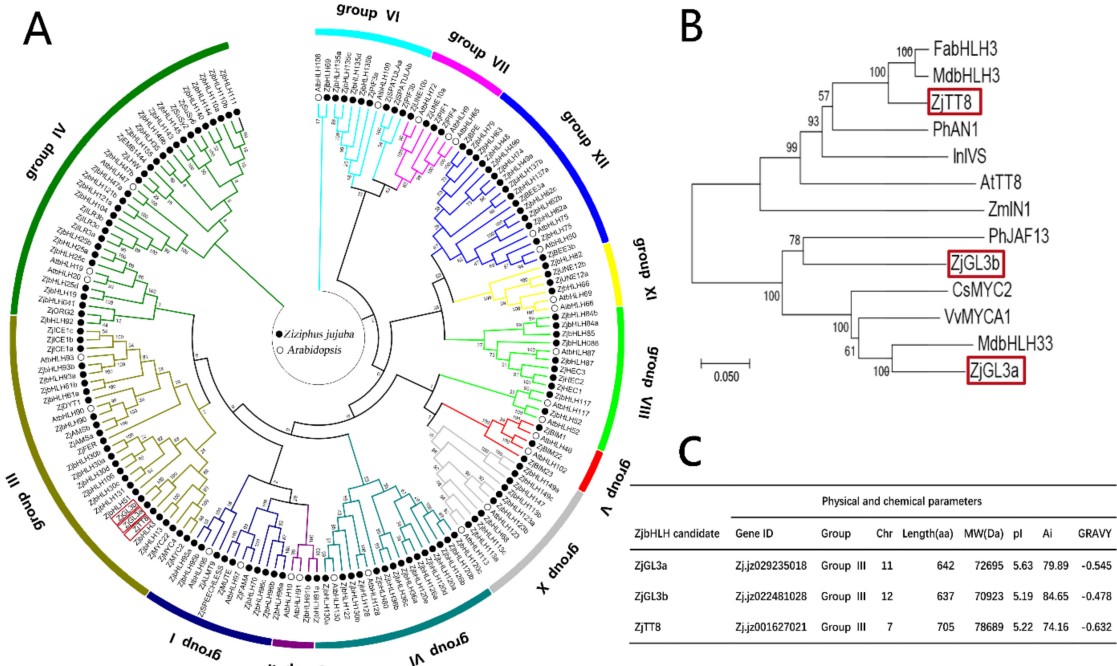

**Figure 1.** (**A**) Phylogenetic analysis of *ZjbHLH* proteins from jujube and Arabidopsis. The filled circles represent the jujube *ZjbHLH* transcription factor and the open circles represent the Arabidopsis *bHLH* transcription factor. (**B**) Phylogenetic analysis of *ZjGL3a, ZjGL3b* and *ZjTT8* with selected anthocyanin pathway bHLH proteins from other plant species. The GenBank accession numbers are: Fragaria x ananassa *FabHLH* (AFL02463.1), Malus domestica *MdbHLH3* (ADL36597.1), Petunia x hybrida *PhAN1* (AAG25927.1), Ipomoea nil *InIVS* (BAE94394.1), Arabidopsis thaliana *AtTT8* (AEE82802.1), Zea mays *ZmIN1* (AAB03841.1), Petunia x hybrida *PhJAF13* (AAC39455.1), Citrus sinensis *CsMYC2* (ABR68793.1), Vitis vinifera *VvMYCA1* (NP_001267954.1), Malus domestica *MdbHLH33* (ABB84474.1) (**C**) Length of protein sequences, molecular weight (MW), theoretical isoelectric point (pI), aliphatic index (Ai), and grand average of hydropathicity (GRAVY) of candidate bHLHs were calculated using the Protparam Expasy tool.

The chromosomal distribution map of jujube bHLH genes was generated on the basis of jujube genome data (Figure 2). The *ZjbHLH* transcription factor family was mapped to the 12 chromosomes of jujube (2n = 2X = 24). The distribution of *ZjbHLH* transcription factors on the chromosomes is not uniform - there are 13 members on chromosomes one and six but only three on chromosome two. The three candidate genes (*ZjTT8*, *ZjGL3a* and *ZjGL3b*) was found on chromosomes 11, chromosomes 12 and chromosomes, respectively.

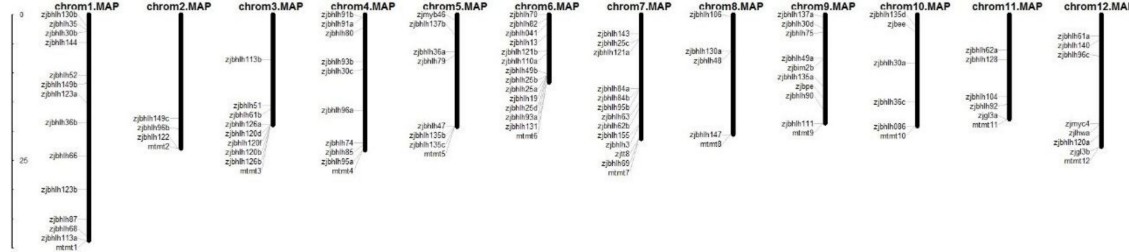

**Figure 2.** Distribution of juvenile bHLH transcription factor members on chromosomes. Chorm1.MAP–Chorm12.MAP indicates 12 chromosomes.

The *ZjbHLH* family protein conserved domain of jujube has a typical Basic-Helix-Loop-Helix structure (Figure 3) [34]. The *ZjbHLH* domain consists of about 60 amino acids, which contain two different functional regions, the C-terminal HLH region and the N-terminal basic amino acid region consisting of 10–15 amino acids. Interestingly, all members had a typical H5-E9-RR13 sequence (His5-Glu9-Arg13) and a few contained a R8-E9 sequence, which were required for specific binding of the target *ZjbHLH* protein to the target DNA. According to the meme-suite website (http://meme-suite.org/), all the bHLH members were subjected to motif analysis. Five motif structures were found (see Figures S2 and S3).

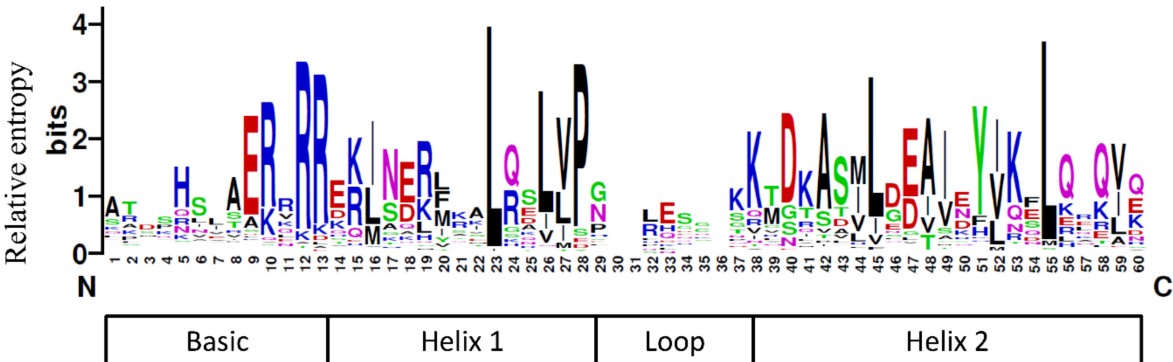

**Figure 3.** Jujube domain of bHLH family protein. Sequence logos of the bHLH domain by the online Gene Structure Display Server; the overall height of each stack represents the degree of conservation at this position, while the height of the letters within each stack indicates the relative frequency of the corresponding amino acids.

*3.2. ZjbHLH Transcription Factor Family Expression Pattern by RNA-Seq Data*

To further confirm the function of *ZjbHLHs* during development and ripening processes in jujube fruit, we explored the expression patterns of each gene using RNA-Seq. As shown in Table S1, a total of 51,073,436–64,668,776 raw reads were generated from the eighteen libraries. After filtration, a total of 49,129,642-62,254,438 clean reads were obtained from the eighteen libraries with an average Raw Q20 and Q30 base rate of nearly 95% and 90%, respectively.

The expression patterns of all *ZjbHLH* transcription factor families of fruit development in the two jujube cultivars, JZ and TLH, were analyzed (Figure 4). We found 71 *ZjbHLH* significantly expressed transcription factors. A heat map was constructed from these 71 significantly expressed transcription factors. At the same time, based on the expression trend, the *ZjbHLH* transcription factors were divided into four groups and the expression pattern of *ZjbHLH* transcription factors was similar in each group. From the heat map, we find that the various members have distinct expression patterns during fruit development. Also, *ZjbHLH* expressions differed between the two cultivars. Interestingly, the expression levels of most *ZjbHLH* transcription factors decreased during maturation. In addition,

*ZjGL3a, ZjGL3b and ZjTT8*, associated with anthocyanin synthesis, have similar expression patterns and are clustered in the same branch.

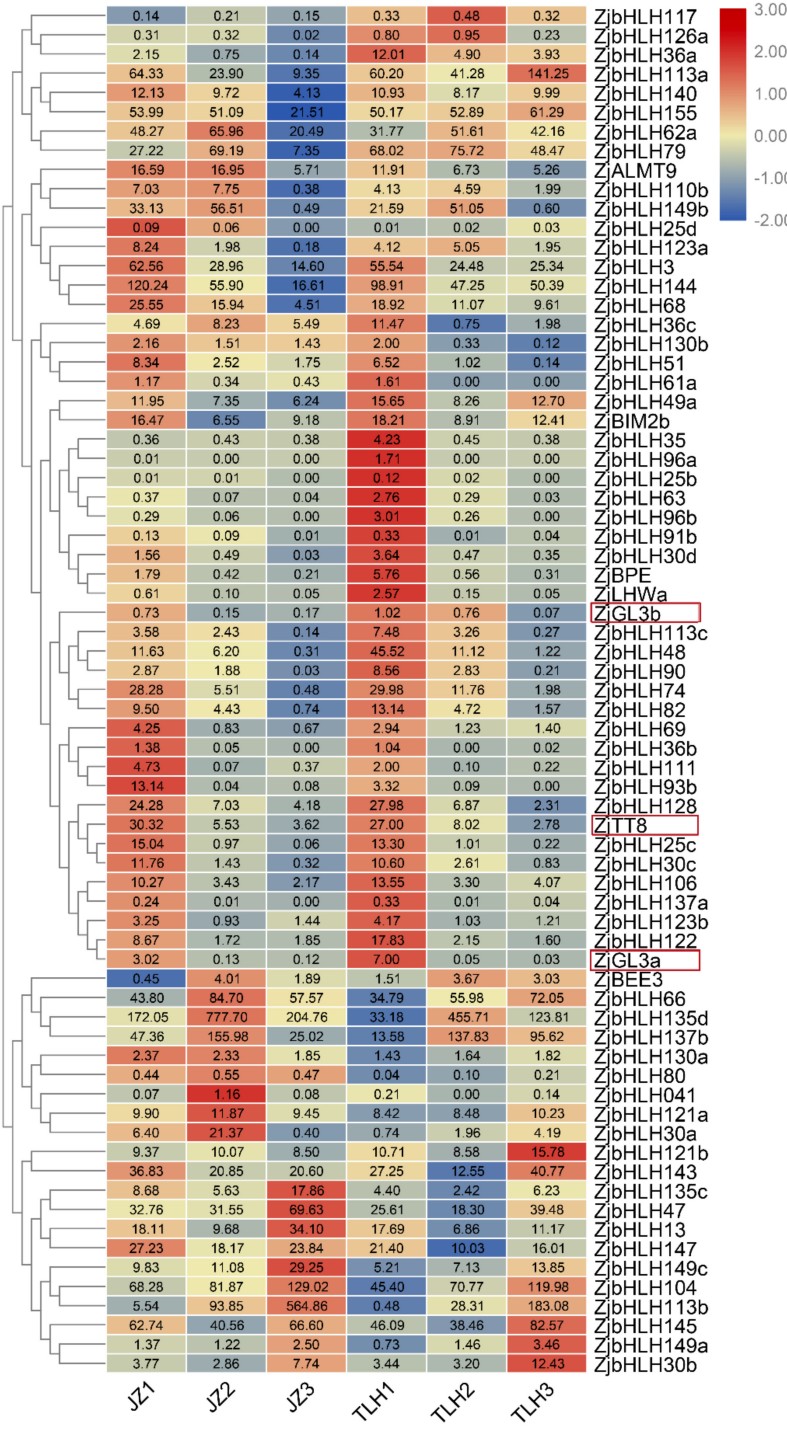

**Figure 4.** Gene expression heatmap of the complete ZjbHLHs in jujube fruits at different developmental stages. Sample names JZ1–TLH3 correspond to days 50, 90 and 110 after anthesis in the jujube cultivars 'Junzao' and 'Tailihong'. The color scale shown at the top represents log2-transformed FPKM (fragments per kilobase of exon per million mapped reads) counts. Original FPKM counts are displayed in the corresponding rectangles. Red indicates high expression and blue indicates low expression. The *ZjbHLHs* with zero expression were eliminated at all developmental stages. The red rectangles indicate three candidate genes for anthocyanin synthesis.

### 3.3. Expression of ZjGL3a, ZjGL3b and ZjTT8 Correlated with Anthocyanin Content

To determine whether the changing expressions of *ZjGL3a*, *ZjGL3b* and *ZjTT8* in jujube fruit correlate with changes in anthocyanin content during development, fruit of JZ and TLH were collected for analysis by the pH differential method and real-time quantitative PCR (Figure 5). In S1, the anthocyanin content in TLH was significantly higher than in JZ. During fruit development and ripening, the anthocyanin contents of both cultivars decreased rapidly. The anthocyanin content of TLH in S3 decreased strongly to 12% of that at S1, while the anthocyanin content in JZ was not detectable in S3. The sequence homology of the data *ZjGL3a*, *ZjGL3b* and *ZjbTT8* with grape and orange was high by evolutionary analysis and their expression levels were consistent with the trend for anthocyanin content. Hence, we questioned if *ZjGL3a*, *ZjGL3b* and *ZjbTT8* might be involved in the regulation of anthocyanin synthesis and if they might also be key regulators of anthocyanin synthesis in jujube. In other fruit species, such as apple and sweet cherry, the expressions of the anthocyanin regulators *MdbHLH3* [26] and *PabHLH3* [35] are closely correlated with gene expression.

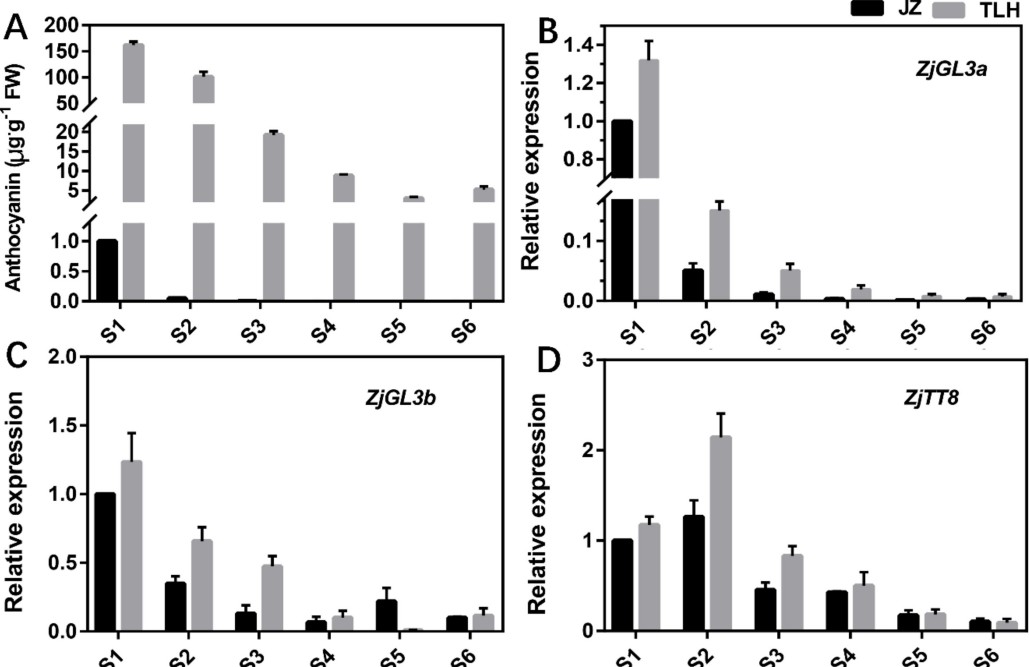

**Figure 5.** Content of anthocyanin and expression patterns of *ZjGL3a*, *ZjGL3b* and *ZjTT8* in the jujube fruit cultivars 'Junzao' and 'Tailihong' at different stages (**A–D**). Error bars indicate SD from three replicates. Developmental stages S1–S6 correspond to 30, 50, 80, 90, 100 and 110 days after anthesis.

## 4. Discussion

Jujube is an important economic species of the Rhamnaceae family. The fruit of jujube is rich in nutrients, which can provide essential amino acids and have strong antioxidant activity. Studying the transcription factor family of jujube is great significance for breeding excellent cultivars and improving fruit quality. However, until now, this TF family has not been reported in jujube. The bHLH transcription factor family has been identified in many species, for example, tomato [21], peach [36], apple [37] and potato [38]. They have many members, various functions, and plays an important role in the growth, metabolism, biosynthesis and stress tolerance of organisms [39–41].

The cluster analysis identified 138 bHLH transcription factors in jujube, which were divided into 12 subfamilies (Figure 1A) based on the reported evolutionary relationships Arabidopsis. The III subfamily of bHLH transcription factors play an irreplaceable role on the regulation of fruit anthocyanins synthesis [24,26]. Interestingly, *ZjGL3a*, *ZjGL3b* and *ZjTT8* were classified into the III subfamily. The *ZjbHLH* domain consists of about 60 amino acids, which contain two different

functional regions, the C-terminal HLH region and the N-terminal basic amino acid region consisting of 10–15 amino acids. And all members had a typical H5-E9-R13 sequence (His5-Glu9-Arg13) and a few contained a R8–E9 sequence, which were required for specific binding of the target *ZjbHLH* protein to the target DNA [42]. Analyzed the conserved domain of the *ZjbHLH* protein sequence showed that the most common amino acids at positions 5, 9 and 13 were histidine, glutamate and arginine, respectively. The analysis of *ZjbHLH* gene expression indicated that the expression levels of *ZjGL3a*, *ZjGL3b*, and *ZjTT8* were consistent with the trend of jujube anthocyanin content. Therefore, *ZjGL3a*, *ZjGL3b* and *ZjTT8* play an important role in anthocyanin synthesis.

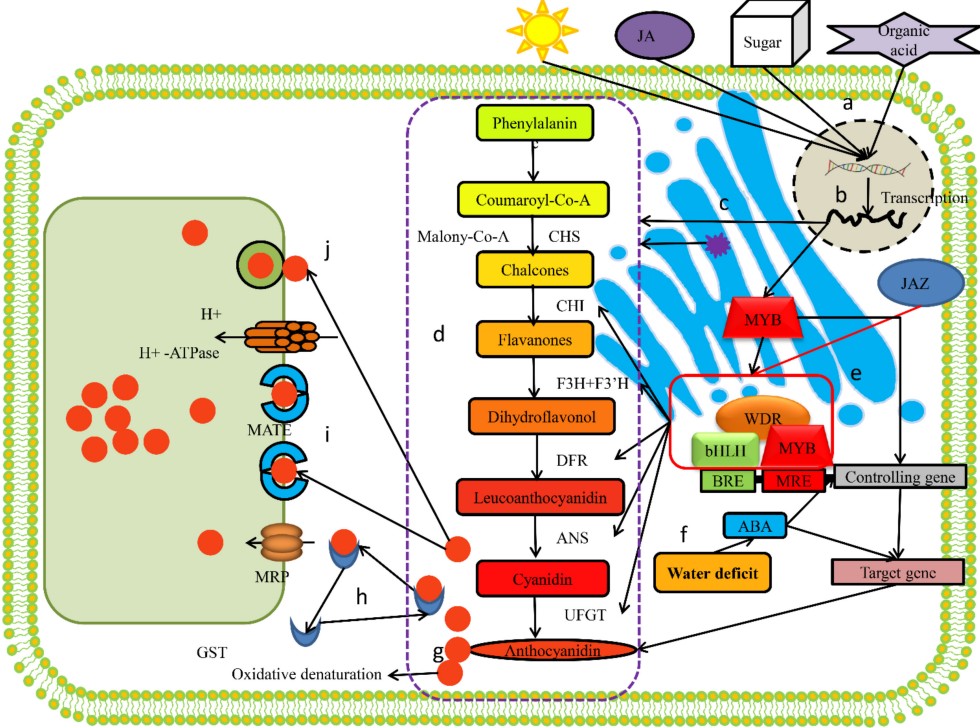

**Figure 6.** Anthocyanin regulatory interaction model. (**a**) Light, hormones, sucrose and organic acids can induce the coding of genes involved in anthocyanin synthesis. (**b**) Transcription. (**c**) Anthocyanin synthesis related genes synthesize related regulatory substances on the endoplasmic reticulum. (**d**) Classical anthocyanin synthesis pathway. (**e**) MYB can bind directly to anthocyanin-related genes and promote anthocyanin synthesis. It can also form complexes with bHLH and WDR to promote the anthocyanin synthesis. bHLH binds to the bHLH- recognition elements (BRE) region of the gene and MYB binds specifically to the MYB- recognition elements (MRE) region to promote downstream responses. (**f**) Plants can synthesize abscisic acid (ABA) in large quantities under water deficit and ABA can then activate the anthocyanin synthesis pathway. (**g**) Anthocyanin oxidative degeneration. (**h**) Anthocyanins are transported to the vicinity of the vacuole with the help of glutathione-S-transferase (GST) and are transported to the inside of the vacuole by the multidrug resistance-associated protein (MRP) transport channel. (**i**) Anthocyanins are transported into the vacuole under the action of the mate-type transporters (MATE) but an H+ concentration gradient, generated by the H+-ATPase pump, is required as a power source. (**j**) Anthocyanins can be engulfed in vacuoles by vesicles.

The MBW complex formed by the combination of MYB, bHLH and WD40 plays a key role in inducing the anthocyanin synthesis pathway (Figure 6). The MBW complex specifically recognizes the MYB- recognition elements (MRE) and the bHLH- recognition elements (BRE). The transcription of the target gene is promoted by binding to a recognition site, upstream of the anthocyanin synthesis-related gene. The synthesis of anthocyanins is a branch of flavonoid synthesis and the synthesis of both substances starts from phenylalanine. Then coumaroyl-Co-A is synthesized from phenylalanine. Coumaroyl-Co-A synthesized chalcones under the action of chalcone synthase (CHS). Chalcones

form flavanones under the catalysis of chalcone isomerase (CHI) and flavanones in flavanone 3-hydroxylase dihydroflavonols are formed by the action of (F3H) and flavonoid 3'-hydroxylase (F3'H), and leucoanthocyanidins are formed by dihydroflavonols under the action of dihydroflavonol 4-reductase (DFR). Subsequently, leucoanthocyanidins form anthocyanidins under the catalysis of anthocyanidin synthase (ANS). Finally, anthocyanidins form anthocyanins under the action of UDP g1ucose-flavonoid 3-O-glucosyltransferase (UFGT). The MBW complex is involved in multiple reaction steps of anthocyanin synthesis and is capable of binding to CHI, F3H, F3'H, DFR, ANS and UFGT and promoting the synthesis of anthocyanins [43]. ZjGL3a, ZjGL3b and ZjTT8 have high homology with bHLH transcription factors involved in anthocyanin synthesis in other fruit species, revealing that ZjGL3a, ZjGL3b and ZjTT8 may play an important role in anthocyanin synthesis of jujube fruits. In addition, bHLH is also used as a messenger of the external signal and can interact with WD40 and MYB to form a complex to promote the efficient synthesis of anthocyanins. There is a close interaction between environment and plant growth, with plants activating or inhibiting various signaling pathways in response to environmental change. Thus, light can activate the MYB transcription factor involved in anthocyanin synthesis, thereby promoting the accumulation of anthocyanins [44,45]. Water deficit during fruit development can enhance the regulation of anthocyanin synthesis by various signaling pathways. The main route may be that plants synthesize ABA activates ABA signaling pathway under drought stress and then promotes anthocyanin synthesis. The expressions of related genes promote fruit ripening and the accumulation of anthocyanins [46,47]. Hormones [27], sugar [48–50], organic acids [51] have also been widely involved in the synthesis, modification and transport of anthocyanins. Hormones, sucrose and organic acids promote the expressions of anthocyanin-related genes by activating the MYB transcription factor family and forming complexes with bHLH and WDR. At the same time, some MYB members are also able to directly regulate the synthesis of related proteins, but the promotion effect is not as strong as the complex.

In general, anthocyanins are synthesized on the Golgi bodies, some are oxidatively denatured and most are eventually transported to the vacuole for storage. There are three general ways to transport anthocyanins into the vacuole [52–54]. (1) Anthocyanin is transported to the vicinity of the vacuole with the help of GST, and then recognized by MRP on the tonoplast and transported into the vacuole; (2) The MATE transmembrane transporter on the vacuolar membrane can be directly transported into the vacuole, which depends on the $H^+$ concentration gradient generated by the $H^+$-ATPase proton pump; (3) Anthocyanins can be directly encapsulated into the vacuole by vesicles.

## 5. Conclusions

We identified and classified 138 *ZjbHLH* transcription factor family members in the jujube genome. Based on genetic analysis, the jujube bHLH transcription factors were divided into 12 subfamilies and found to be located on 12 chromosomes. The *ZjGL3a*, *ZjGL3b* and *ZjTT8* transcription factors associated with anthocyanin synthesis are all classified in group III. *ZjGL3a*, *ZjGL3b* and *ZjTT8* are highly homologous to the anthocyanin synthesis genes in grape and orange, and their expression levels are consistent with anthocyanin content. This shows they are important transcription factors involved in anthocyanin regulation in jujube fruits. The expression patterns of bHLH transcription factors in the jujube cultivars TLH and JZ were examined during development. It was found the expression levels of most transcription factors showed a downward trend. The content of total anthocyanins in TLH and JZ gradually decreased during fruit development. The relative expression levels of *ZjGL3a*, *ZjGL3b* and *ZjTT8* transcription factors associated with anthocyanin synthesis also showed a downward trend. An anthocyanin synthesis-transport model centered on the WDR/bHLH/MYB complex was established to provide a reference for further understanding of the regulation of anthocyanin synthesis.

**Supplementary Materials:** The following are available online at http://www.mdpi.com/1999-4907/10/4/346/s1, Figure S1: The pictures of two jujube fruit developments. Figure S2: Jujube ZjbHLH transcription factor protein motif structure. Figure S3: Conserved motif analysis of Jujube of ZjbHLH gene family. Table S1: Statistic Reads of RNA-Seq data.

**Author Contributions:** Q.S. designed the experiments, wrote and revised the manuscript. X.L. (Xi Li) and J.D. helped revised manuscript. X.L. (Xingang Li) designed the experiments, discussed results, and revised the manuscript.

**Funding:** This work was supported by The National Key Research and Development Program of China (grant number 2018YFD1000607).

**Acknowledgments:** The authors are grateful to Huang Dong and Gu Bao for their suggestions for the experiment.

**Conflicts of Interest:** The authors declare no conflict of interest.

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
