# Peer review of "Anthocyanin Synthesis and the Expression Patterns of bHLH Transcription Factor Family during Development of the Chinese Jujube Fruit (Ziziphus jujuba Mill.)"

_forests, doi:10.3390/f10040346_

Reviewer 1 Report

The authors evaluated two jujube cultivars to identify and analyze the bHLH transcription factors and investigate their changing expressions during fruit development. They identified and classified 138 ZjbHLH transcription factor family members. Based on genetic analysis, the jujube bHLH transcription factors were divided into 12 subfamilies and were located on 12 chromosomes. The ZjGL3a, ZjGL3b and ZjTT8 256 transcription factors associated with anthocyanin synthesis are all classified in group III.

Introduction and Materials and Methods are well written. But results and discussion are very poorly written, explained. Discussion part is totally wrong. Authors need to discuss their results, instead of providing general information about anthocyanin synthesis.

Following are comments for changes.

Line 15:  Define bHLH? Before you use the abbreviation, first define it.

Line 16: what do you mean by complex? Is it a complex structure or network? Mention that.

Line 16: Define WD40 and MYB? Before you use the abbreviation, first define it.

Line 17: Define ZjbHLH?

Line 19: what about other genes?

Line 20: Define AtbHLHs?

Line 39: Use comma, “Here, they

Line 47: use “The”. The bHLH has been shown to be involved in numerous biological processes,

Line 66: use “was”, instead of “is”. Our aim was to.

Line 72: why did you use these two cultivars?

Line 74: how many fruit samples?

Line 92: add the website for TBTOOLS software.

Paragraphs are not in order in the materials and methods. Reorder the paragraphs in the materials and methods in this order: 2.1, 2.4, 2.5, 2.6, 2.2, 2.3

Line 154: Need more detail in figure 2 caption. Mention about 12 chromosomes in the caption.

You did not use figure 2 in the text. Mention it in the text.

Line 159: Again, need more detail in figure 3 caption. Explain all different colors and axis.

Line 170: use comma after cultivars “cultivars, JZ and TLH,”.

You gave figure 5, but there is no figure 4.

Line 180: Need more detail in figure 5 caption. Mention what are three red rectangles? Explain what are those numbers in this figure 5? How are we relating these number for changing gene expressions?

You did not use figure 5 in the text. Mention it in the text.

You did not use figure 6 in the text. Mention it in the text.

Use same X-axis numbers in figure 6 for ZjGL3a, ZjGL3b, and ZjTT8.

Line 201: Discussion:

There is no discussion about your results. In the discussion section, you need to discuss your results, not general information about anthocyanin synthesis.

In the discussion, you are just talking about how important transcription factor involved in anthocyanin synthesis. This is not how you right discussion. Rewrite your whole discussions.

Author Response

Responses to reviewer

Comments: Introduction and Materials and Methods are well written. But results and discussion are very poorly written, explained. Discussion part is totally wrong. Authors need to discuss their results, instead of providing general information about anthocyanin synthesis.

Line 15: Define bHLH? Before you use the abbreviation, first define it.

[Response] I’m sorry for this a mistake, bHLH has been defined in line 15 of my corrected manuscript.

Line 16: what do you mean by complex? Is it a complex structure or network? Mention that.

[Response] It is our mistake, it is a ternary structure complex, and we have mentioned it line 18 of my corrected manuscript.

Line 16: Define WD40 and MYB? Before you use the abbreviation, first define it.

[Response] You are right, the full name should be contain because they appear for the first time in our manuscript, we have added their full name in corrected manuscript. Please see in line 18.

Line 17: Define ZjbHLH?

[Response] Corrected. Please see in line 19.

Line 19: what about other genes?

[Response] A total of 145 ZjbHLHs sequences was screened from the JZ genome and the conserved domain motifs of the sequence was analyzed by Pfam online site (http://pfam.xfam.org/) to remove the incomplete sequence of the conserved domain. Finally, 138 ZjbHLH transcription factor family members were obtained.

Line 20: Define AtbHLHs?

[Response] Corrected. Please see in line 22.

Line 39: Use comma, “Here, they

[Response] Corrected. Please see in line 43.

Line 47: use “The”. The bHLH has been shown to be involved in numerous biological processes,

[Response] Corrected. Please see in line 68.

Line 66: use “was”, instead of “is”. Our aim was to.

[Response] Corrected. Please see in line 50.

Line 72: why did you use these two cultivars?

[Response] That's why I chose the two jujube cultivars ‘Junzao’ and ‘Tailihong’ in this study. The first was the commercial cultivar ‘Junzao’ whose skin color changes from green to white during early ripening and later turns red. The second was a color mutant ‘Tailihong’, in which the skin color changes from purple-red in young fruit to yellow at the beginning of ripening and then to red at full ripening. Please see the pictures of the two jujube cultivars in supplementary figure 1.

Line 74: how many fruit samples?

[Response] Thank you very much for your attention. At least 20 fruits of each sample and three replicates were performed. Please see in line 77.

Line 92: add the website for TBTOOLS software.

Paragraphs are not in order in the materials and methods. Reorder the paragraphs in the materials and methods in this order: 2.1, 2.4, 2.5, 2.6, 2.2, 2.3

[Response] I’m sorry for this a mistake. The website and reference have added it in this section. Please see in line 121. And, according to the reviewer’s suggestions, the paragraphs in the materials and methods have been reordered. 

Line 154: Need more detail in figure 2 caption. Mention about 12 chromosomes in the caption.

You did not use figure 2 in the text. Mention it in the text.

[Response] According to the reviewer’s suggestions, we have added more detail in figure caption and mentioned it in the text. (Line 185-194) 

Line 159: Again, need more detail in figure 3 caption. Explain all different colors and axis.

[Response] According to the reviewer’s suggestions, we have added more detail in figure caption. Please see in line 197-209.

Line 170: use comma after cultivars “cultivars, JZ and TLH,”.

[Response] Corrected. Please see in line 218.

You gave figure 5, but there is no figure 4.

[Response] I’m sorry to this a mistake, we have corrected.

Line 180: Need more detail in figure 5 caption. Mention what are three red rectangles? Explain what are those numbers in this figure 5? How are we relating these number for changing gene expressions?

[Response] Thank you very much for your attention. We have added more detail in figure caption. The color scale shown at the top represents log2-transformed FPKM (fragments per kilobase of exon per million mapped reads) counts. Original FPKM counts are displayed in the corresponding rectangles. Red indicates high expression and blue indicates low expression. The ZjbHLHs with zero expression were eliminated at all developmental stages. The red rectangles indicate three candidate genes for anthocyanin synthesis. (Line 230-234)

You did not use figure 5 in the text. Mention it in the text.

[Response] Corrected.

You did not use figure 6 in the text. Mention it in the text.

[Response] Corrected.

Use same X-axis numbers in figure 6 for ZjGL3a, ZjGL3b, and ZjTT8.

[Response] Thanks for your comments. There are two reasons chose three figures for ZjGL3a, ZjGL3b, and ZjTT8. First, it is difficult for us to put them in a picture because of there are two jujube cultivars (Junzao and Tailihong), three genes (ZjGL3a, ZjGL3b, and ZjTT8) and six developmental stages. Second, They could only represent a trend of genes because of ZjGL3a, ZjGL3b, and ZjTT8 were relative expression., therefore, we chose three figures for ZjGL3a, ZjGL3b, and ZjTT8.

Line 201: Discussion:

There is no discussion about your results. In the discussion section, you need to discuss your results, not general information about anthocyanin synthesis.

In the discussion, you are just talking about how important transcription factor involved in anthocyanin synthesis. This is not how you right discussion. Rewrite your whole discussions.

[Response] Thank you very much for your important assessment. We have thoroughly rewritten this paragraph according to your suggestions. Please see in 4. Discussion.

Reviewer 2 Report

The paper entitled “Anthocyanin synthesis and the expression patterns of 2bHLH transcription factor family during development of Chinese jujube fruit (Ziziphus jujuba Mill.)”  provides for the first time the regulatory pathway model for anthocyanin synthesis for Ziziphus jujuba.

Below you can find some minor comments from my point of view,  which are more targeted to language usage.

Comments:

Line 36: Please rephrase the sentence like ‘This spesies is in high demand due to its rich flavor and high nutritional 36 value in many Asian countries’

Line 37:  Replace “has shown jujube to be” with “has demonstrated that jujube is”

Line 62: Rephrase the whole sentence

Line 67: Erase “It is hoped” with “Our aim is that these findings will…”

Line 79: Erase “was carried out”

Line 121: Erase” of the conserved domain”

Line 125: Replace contain with contained

Line 207: Rephrase “ABA under drought stress, and also activates the ABA signaling pathway with final promotion of anthocyanin synthesis.”

Line 241: erase “and the like”

Author Response

Responses to reviewer:

Line 36: Please rephrase the sentence like ‘This species is in high demand due to its rich flavor and high nutritional 36 value in many Asian countries’

[Response] Thank you very much for your attention. We have rephrased the sentence. (Line 37-38)

Line 37:  Replace “has shown jujube to be” with “has demonstrated that jujube is”

[Response] Corrected.

Line 62: Rephrase the whole sentence

[Response] Thank you very much for your attention. We have rephrased the sentence. Please see in line 61-62 in corrected manuscript.

Line 67: Erase “It is hoped” with “Our aim is that these findings will…”

[Response] Corrected.

Line 79: Erase “was carried out”

[Response] Corrected.

Line 121: Erase” of the conserved domain”

[Response] Corrected.

Line 125: Replace contain with contained

[Response] Corrected.

Line 207: Rephrase “ABA under drought stress, and also activates the ABA signaling pathway with final promotion of anthocyanin synthesis.”

[Response] Corrected. Please see in line 313-314.

Line 241: erase “and the like”

[Response] Corrected.

Reviewer 3 Report

Please correct the suggested comments

Author Response

Responses to reviewer:

Line 73: Northwest A&F University Give the full form of these abbreviations.

[Response] I’m sorry for this a mistake, Northwest A&F University’ has been given the full form in my corrected manuscript. (Line 78)

Line 74-75: ‘Fruit samples were harvested at six developmental stages (S1…S6) on days 30, 50, 80, 90, 100 and 110 74 after anthesis (DAA), respectively.’ How many fruits per dev. stage used and from how many plants, please elaborate. Please add the pictures of the fruit developmental stages in supplementary figure.

[Response] Thank you very much for your constructive comments. At least 20 fruits of each sample from three jujube trees were harvested (line 79). And, according to the reviewer’s suggestions, we have added the he pictures of the fruit developmental stages in supplementary figure 1.

Line 75: ‘The fruit skins (about 2 mm thick)..’ fruit skin removed from all fruit body or randomly removed? please elaborate.

[Response] The fruit skins (about 2 mm thick) were removed with a domestic vegetable peeler randomly. Please see in line 82.

Line 88. add full stop.

[Response] Corrected.

Line 92. ‘The heatmap 91 was generated with TBTOOLS softwareadd ref.

[Response] I’m sorry for this a mistake. The website and reference have added it in this section. Please see in line 121.

Line 95. ‘Samples (about 0.5 g) of jujube fruit…….’ From how many sample the 0.5 g skin sample prepared.

[Response] Thank you! At least 20 fruits of each sample. Please see in line 79 in our revised manuscript.

Figure 1C Font in table unreadable and unclear use same font in figure A and B.

[Response] Thank you very much for your attention. We have redrawn the figure. Please see the figure 1.

Figure 2 What is this scale and its unit? please write elaborately in the figure legend.

[Response] I’m sorry for this a mistake, the scale and unit of the figure 2 was 25 cM. Please see figure 2.

Figure 3 Remove the grid lines in the figure and write the figure legend elaborately.

[Response] Thank you very much. We have redrawn the figure. Please see the figure 1.

Figure 5 Font in the figure is not clear and use the same font, it looks broken font.

[Response] Sorry, we regret to hear for our so basic errors. We have redrawn the figure. Please see the figure 3.

Figure 7 Many of the abbreviation in the figure don’t have the full forms. please add the full forms of abbreviation used in the figures into figure legends.

[Response] You are right, the full name have been added in our revised manuscript (Lines 282-289).

Line 250. ‘The MATE transmembrane transporter.’ give full forms.

[Response] In our corrected manuscript, we have given the full forms ‘mate-type transporters’, please see in line 288.

Line 227. Reference Cross check the reference orders, journal name abbreviations and S.N must be italics.

[Response] I’m sorry to this a mistake, the reference orders have been checked, the journal name abbreviations and S.N have been changed to italics. 

Line 280. cite correctly journal name PLOS.

[Response] Corrected.

Line 284. ‘Spanish jujube (Ziziphus jujuba Mill.)’ italics.

[Response] Corrected.

Line 288. ‘Trends Plant Sci’ give full forms.

[Response] Corrected. The full forms have been given.

Line 298. ‘BMC plant Biol’ give full forms.

[Response] Corrected. The full forms have been given.

Round  2

Reviewer 1 Report

Authors made all changes as suggested. 

Line 38: What is 36 here? I think you copied and pasted other reviewer suggested. Because he didn’t delete the 36 from his sentence. That’s why you still have. 

Author Response

Line 38: What is 36 here? I think you copied and pasted other reviewer suggested. Because he didn’t delete the 36 from his sentence. That’s why you still have.

[Response] Sorry, we regret to hear for our so basic errors. We have deleted it.